# Improving private tuberculosis patient notifications in Akot Tuberculosis Unit of Vidarbha region of India: A Plan-Do-Study-Act (PDSA) based approach for enhanced public health reporting

Umesh Kawalkar[1]◐*, Manish Sharma[2], Abhay Gaidhane[3]◐, Amar Mankar[3], Snehal Chavhan[4], Mahesh Puri[5], Priti Kogade[2], Anshu Singh[6], Zahiruddin Quazi Syed[3]

1 Department of Community Medicine, Government Medical College, Washim, Maharashtra, India, 2 Public Health Department, Akola, Maharashtra, India, 3 Department of Community Medicine, Datta Meghe Institute of Higher Education and Research, Sawangi (M), Wardha, Maharashtra, India, 4 ICMR- National Institute of Virology, Pune, Mumbai, India, 5 ESIC Medical College, Hyderabad, India, 6 Department of Community Medicine, Government Medical College, Akola, Maharashtra, India

◐ Umesh Kawalkar and Abhay Gaidhane contributed equally to this work and share first authorship.
* umeshkawalkar01@gmail.com

## Abstract

India bears the world's largest tuberculosis (TB) burden, and under-reporting from private-sector providers continues to hinder elimination goals. In the Akot Tuberculosis Unit (TU) of Maharashtra, private notifications remained disproportionately low despite extensive private healthcare presence. This study implemented a structured quality-improvement (QI) model using Plan–Do–Study–Act (PDSA) cycles to enhance TB case notifications from private providers. A mixed-method quality-improvement study was conducted from August to December 2023. Four iterative PDSA cycles were implemented targeting private practitioners, pharmacists, and laboratory technicians. Interventions included sensitization workshops, weekly WhatsApp-based digital outreach, and personalized follow-up visits by Public–Private Mix (PPM) coordinators. Quantitative data were extracted from the *Ni-kshay* platform and validated through Tuberculosis Unit records. Private notification trends (2018–2024) were analysed using segmented regression to assess temporal trends, while qualitative insights from key informant interviews (n = 6) explored barriers and enablers influencing notification practices. A total of 90 stakeholders participated (45 practitioners, 30 pharmacists, and 15 laboratory technicians). Private TB notifications increased from **16 of 90 total cases (17.8%)** during January–July 2023–**63 of 121 cases (52.1%)** during November–December 2023, reflecting increase during the post-intervention period. The interrupted time-series analysis demonstrated a positive post-intervention trend in private TB notifications (β = +50.5, p = 0.032). Qualitative findings revealed that personalized digital mentoring and hands-on technical support

**Data availability statement:** All anonymized data and documentation of PDSA cycles are provided as supplementary files accompanying this paper.

**Funding:** The authors received no specific funding for this work.

**Competing interests:** The authors declare that they have no competing interests.

helped address barriers such as Ni-kshay login issues, misconceptions about notification responsibility, and perceived workload. The implementation of a structured PDSA-based quality-improvement approach was associated with improvements in private-sector TB notifications in a previously low-performing tuberculosis unit. The findings suggest that integrating digital engagement, capacity building, and supportive supervision within existing NTEP structures may strengthen TB surveillance and private-sector engagement. These insights may inform scalable strategies to improve TB notification systems in similar programmatic settings.

## Introduction

Tuberculosis (TB) remains one of the most pressing public health challenges globally, with India bearing the highest burden of the disease [1]. Despite significant advancements in diagnosis and treatment, the World Health Organization estimates that nearly 2.4 million TB cases were reported in India in 2023 alone, with many cases either underreported or unreported [2]. The private healthcare sector plays a crucial role in TB case management, as a significant proportion of patients initially seek care from private practitioners, pharmacies, or diagnostic laboratories. However, despite their importance, private-sector contributions to TB case notifications have historically been inconsistent due to a lack of awareness, technical challenges, and inadequate follow-up mechanisms [3–5]. The National Tuberculosis Elimination Program (NTEP) in India has prioritized engaging private healthcare providers to bridge this gap in TB case detection and notification. Policies such as mandatory TB notification and incentives for private practitioners have improved notification rates over the years [6,7]. Nationally, India's private health sector has progressively become a major contributor to TB notifications under the National TB Elimination Programme (NTEP). In India, private-sector notifications contribute approximately one-third (~30–33%) of total TB notifications, as reported in the national India TB Reports [7]. This rise reflects the cumulative impact of national initiatives such as the *Ni-kshay* platform, Public–Private Mix (PPM) schemes, and incentive-based engagement models. To improve TB surveillance, the Government of India launched "Ni-kshay," a national digital system developed by the Central TB Division to record, monitor, and track TB cases in real time. The word "Ni-kshay" is derived from Sanskrit, where *"Ni"* means *free from* and *"Kshay"* means *tuberculosis,* signifying the goal of a *TB-free India.* However, the situation at the sub-district level remains heterogeneous. For example, in the Akot Tuberculosis Unit (TU) of Maharashtra which serves a population of about 280 000 and includes 52 IMA doctors, 65 NIMA practitioners, 14 private labs, and 60 chemists only 16 private TB patient notifications were recorded between January and July 2023. This disparity underscores the need for targeted, local quality-improvement approaches to enhance private-sector engagement.

This low rate highlights systemic challenges such as poor coordination between public health systems and private stakeholders, limited technical knowledge, and low awareness about notification protocols. Efforts to address these gaps have

increasingly focused on innovative, structured approaches. The Plan-Do-Study-Act (PDSA) cycle has emerged as a widely adopted framework for quality improvement in healthcare systems. By enabling iterative testing and refinement of interventions, the PDSA cycle allows for tailored solutions to complex challenges [8,9]. Previous studies in India and other high TB-burden countries have demonstrated the effectiveness of PDSA-based approaches in enhancing health outcomes, particularly in improving case detection and notification rates in underserved populations [10]. This study builds on this foundation, aiming to improve private-sector TB notifications in Akot TU through a series of targeted interventions implemented within the PDSA framework. By focusing on sensitizing private practitioners, pharmacists, and laboratory technicians, and leveraging digital tools like WhatsApp for communication and follow-up, this study seeks to address the unique challenges of this region. Furthermore, the study aligns with the National Tuberculosis Elimination Program (NTEP) goals, which emphasize private-sector integration. The findings from this initiative have the potential to inform scalable models for TB notifications in other regions facing similar challenges, contributing to the broader goal of TB elimination.

## Methodology

### Study design

This study employed a quality improvement methodology using the Plan-Do-Study-Act (PDSA) cycle [9]. This quality-improvement initiative adopted a mixed-methods quasi-experimental design, integrating process-improvement (PDSA) cycles with both quantitative outcome tracking and qualitative feedback from stakeholders to guide iterative refinement of the intervention. The PDSA framework was chosen to facilitate systematic testing and implementation of interventions aimed at increasing private TB patient notifications in the Akot Tuberculosis Unit (TU) of Akola district, Maharashtra. This iterative approach enabled continuous evaluation and refinement of strategies to achieve the desired outcomes. This study was conducted as a programmatic quality-improvement initiative embedded within routine National Tuberculosis Elimination Programme (NTEP) activities, and the evaluation reflects real-world programme implementation within the district tuberculosis control system.

### Study area and population

The study was conducted in the Akot Tuberculosis Unit (TU) of Akola district, Maharashtra, which caters to a population of approximately 280,000. Participants were selected using a purposive sampling technique between 1 March and 30 April 2023. The focus was on private healthcare stakeholders including private practitioners, pharmacists, and laboratory technicians who were practicing within the jurisdiction of Akot TU and were not actively engaged in TB notification prior to the intervention. To ensure completeness and minimize selection bias, a comprehensive mapping of all private healthcare facilities in the TU was undertaken. Akot TU has a dense private health network comprising 52 Indian Medical Association (IMA) doctors, 65 National Integrated Medical Association (NIMA) practitioners, 14 private laboratories, and 60 pharmacies or chemists, accounting for a total of 191 private facilities. Mapping was conducted jointly by the District Tuberculosis Officer (DTO) and the Public–Private Mix (PPM) Coordinator using multiple data sources, including the *Ni-kshay* provider database maintained under the National TB Elimination Programme (NTEP), the Clinical Establishment Registry available with the District Health Office, membership lists from local medical associations (IMA and NIMA), and local directories of chemists and diagnostic laboratories verified through field visits. Of these 191 mapped facilities, 118 (61.7%) were already registered on the *Ni-kshay* platform, while 73 (38.3%) were identified through external sources and verified through door-to-door visits by the PPM Coordinator. These additional facilities were sensitized and subsequently linked to *Ni-kshay* during the intervention period. This multi-source triangulation ensured that all known private healthcare providers within Akot TU were captured, thereby improving coverage and reducing the possibility of exclusion or selection bias. The inclusion criteria comprised healthcare providers who were likely to encounter presumptive TB cases during routine practice but had a history of underreporting or non-reporting on *Ni-kshay*. Stakeholders (n: 12) who had already been regularly

notifying TB cases were excluded to focus efforts on those with the highest potential for change. The final list of eligible participants was compiled in consultation with the DTO and PPM Coordinator, ensuring efficient resource utilization and alignment with the Quality Improvement (QI) framework adopted for this study. Although the intervention activities primarily targeted under-reporting providers identified through this mapping exercise, notification outcome data were obtained from the Ni-kshay surveillance system and therefore include notifications from all private healthcare providers operating within the TU jurisdiction.

The exclusion of regularly notifying providers was intentional to maintain the quality-improvement focus on underperforming stakeholders. Regular notifiers were defined as those who had reported at least one confirmed TB case in three or more of the four preceding quarters, as verified through *Ni-kshay* data. In contrast, under-reporting providers had no or minimal notifications during the same period. Including consistent notifiers would have limited the sensitivity of the intervention to detect meaningful behavioural change, as their reporting practices were already compliant with NTEP standards. This classification was finalized in consultation with the District Tuberculosis Officer (DTO) and Public–Private Mix (PPM) Coordinator based on district-level performance records.

## Baseline assessment

A baseline review of *Ni-kshay* data (January–July 2023) showed only 16 private-sector TB patient notifications out of 90 total notifications (17.8%). The assessment included both quantitative and qualitative components to identify operational gaps and behavioural barriers. Quantitative data were extracted from *Ni-kshay* and verified against TU monthly performance registers. To complement the quantitative assessment, a qualitative inquiry using Key Informant Interviews (KIIs) was undertaken to identify behavioural and systemic factors affecting private-sector notification. A total of six KIIs were conducted with the DTO, PPM Coordinator, Senior Treatment Supervisor, two private practitioners, and one pharmacist. Participants were selected purposively to capture perspectives from programme managers and front-line private providers. Semi-structured guides explored awareness of mandatory TB notification, perceived technical and logistic barriers in *Ni-kshay* use, communication channels with the TB programme, and suggestions for improvement. Each interview lasted 30–45 minutes, was conducted in Marathi or Hindi, and documented through detailed field notes following verbal consent.

Baseline notification data obtained from the *Ni-kshay* platform and verified with the District TB Office indicated that Akot TU reported a total of 312 TB patient notifications in 2023, including both public and private sources. Of these, only 16 cases (5.1%) were contributed by private healthcare providers during the baseline period (January–July 2023). This proportion highlights the extent of under-reporting from the private sector before implementation of the quality-improvement intervention. The baseline period (January–July 2023) was selected because it represents the immediate operational context preceding implementation of the PDSA-based intervention, thereby providing a programmatic reference point for evaluating subsequent changes.

## Interventions

The intervention implemented to improve private TB notifications in Akot TU was multi-faceted, comprising capacity-building workshops, digital outreach, and personalized follow-ups. Three Sensitization workshops were conducted for private practitioners, pharmacists, and laboratory technicians, focusing on the legal requirements of TB notification, use of the Ni-kshay platform, and practical case-based demonstrations. These sessions were interactive and supported with printed educational materials. In parallel, a structured digital engagement strategy was deployed using WhatsApp, through which weekly messages including simplified reporting protocols, video demonstrations, and motivational content were disseminated to stakeholders. To reinforce learning and support implementation, the PPM coordinator conducted monthly follow-ups with non-notifying providers to address individual challenges such as technical issues with Ni-kshay, misconceptions about notification, and workload concerns. This adaptive, mixed-mode approach allowed continuous feedback and refinement during each PDSA cycle, ensuring better alignment with provider needs and maximizing the effectiveness of the intervention.

## Data collection

Data collection focused on both process and outcome measures to evaluate the impact of the interventions. Process measures included the number of stakeholders sensitized through workshops, the volume of WhatsApp messages sent and received, and the frequency of follow-ups conducted by PPM coordinators. Outcome measures assessed the increase in private TB notifications. All notification data were extracted from the Ni-kshay national tuberculosis surveillance platform and cross-verified with Tuberculosis Unit programme records to ensure completeness and accuracy. Data were gathered through attendance records, digital communication logs, and the Ni-kshay platform, ensuring a comprehensive evaluation of the implemented strategies. Qualitative data were collected between March and October 2023 as part of routine implementation and evaluation of the PDSA cycles. Data collection was conducted by the authors AM and PK in collaboration with the Public–Private Mix (PPM) Coordinator, both of whom were involved in programme implementation and trained in qualitative data collection. The qualitative component was designed to capture contextual, behavioural, and operational factors influencing private-sector TB notification.

Key informant interviews were conducted using purposive sampling to capture perspectives of stakeholders directly involved in or affected by private TB notification processes. Interviewees included programme managers and frontline private-sector providers. A total of six KIIs were conducted during the baseline phase, including the District Tuberculosis Officer, Public–Private Mix Coordinator, Senior Treatment Supervisor, two private medical practitioners, and one pharmacist. Interviews explored awareness of mandatory TB notification, experiences with the Ni-kshay platform, perceived barriers to notification, and suggestions for improving engagement. Interviews lasted 30–45 minutes, were conducted in Marathi or Hindi, and were documented through detailed field notes following verbal informed consent.

Observational data were collected as part of routine programme implementation during sensitization workshops, field visits, and follow-up interactions with private healthcare providers. These observations were conducted by the Public–Private Mix (PPM) Coordinator and the research team and documented as structured field notes. Observations focused on operational and implementation-related aspects, including provider interaction with the Ni-kshay platform, documentation practices, common reporting errors, time constraints during outpatient consultations, and responsiveness to programme support. Observational data were used to contextualize and triangulate findings from key informant interviews.

## Implementation phases

The study was conducted in four iterative PDSA cycles [11]:

**1. Plan**:

- Identify barriers to private notifications and design tailored interventions.
- Develop workshop content and digital communication templates.

**2. Do**:

- Implement interventions over a three-month period.
- Conduct workshops and initiate digital outreach.

**3. Study**:

- Monitor key metrics, including participation in workshops and changes in notification rates.
- Collect feedback from stakeholders to identify operational challenges.

**4. Act**:

- Refine interventions based on insights from the study phase.
- Scale up successful strategies while addressing identified gaps.

## Data analysis

Process indicators such as the number of stakeholders sensitized through workshops, engagement with WhatsApp messages, and monthly follow-ups conducted by the Public–Private Mix (PPM) Coordinator were analysed using descriptive statistics (frequencies and percentages) to assess the reach and engagement of the interventions. The primary outcome measure was the number of private TB patient notifications recorded on the *Ni-kshay* platform. Baseline data (January–July 2023) were compared with post-intervention data (November–December 2023), and the percentage change in private TB patient notifications was calculated to describe changes in notification patterns following the intervention. January–July 2023 was selected as the baseline period because it immediately preceded implementation of the PDSA-based intervention (August 2023) and therefore represents the operational context immediately prior to intervention initiation. The intervention was implemented between August and October 2023 through iterative PDSA cycles, and the post-intervention period (November–December 2023) therefore represents the first complete operational phase following implementation of the intervention activities. An interrupted time-series (ITS) analysis was performed using annual private tuberculosis notification data from 2017 to 2024 to illustrate longer-term trends in private TB notifications within the TU. Annual data were used for this analysis because consistent monthly notification data were not available for the entire historical period (2017–2022). The years 2017–2022 were considered the pre-intervention period.

The PDSA-based intervention was implemented in 2023, which constituted the interruption point. Segmented regression was applied using a linear model for annual counts, estimating: (i) the baseline secular trend prior to the intervention, (ii) the immediate level change at the time of intervention introduction, and (iii) the change in post-intervention trend (slope). Following recommended ITS parameterization, the post-intervention trend term was defined as time elapsed since intervention onset, ensuring that the post-intervention slope was centred at zero at the point of intervention introduction and thereby avoiding biased estimation of level change. This approach follows best-practice guidance for segmented regression in ITS analyses. The analysis period included years affected by the COVID-19 pandemic (2020–2021), during which TB detection and notification were disrupted nationally; these effects were not modelled explicitly and are considered contextual influences on observed trends. Given the absence of month-wise notification data and population denominators, the analysis was performed on annual counts rather than rates, and Poisson models with population offsets were not used. The ITS analysis was therefore interpreted as assessing associations in trends and levels, rather than precise rate changes. All analyses were conducted using R software (version 4.3.2).

Interview and observational field notes were analysed together using an inductive thematic approach to allow triangulation between reported experiences and observed implementation practices. Coding and theme generation were performed using R-based qualitative data analysis (RQDA), a free, open-source qualitative data analysis package integrated within R. Codes were developed iteratively to capture recurring patterns, and emerging themes were refined through repeated discussion and validation. The coding framework and final themes were independently verified by the corresponding author to ensure analytical consistency and credibility.

## Ethical considerations

Ethical approval for the study was obtained from the Institutional Ethics Committee of Government Medical College, Akola (Reference No. 20/2023, dated 30/01/2023). Participation in the workshops and digital outreach initiatives was entirely voluntary. All participants were informed about the purpose of the study, and verbal informed consent was obtained prior to their involvement in sensitization sessions, interviews, or follow-up interactions which was approved by Institutional Review Board considering the nature of the intervention, the minimal risk involved, and the feasibility constraints in routine field operations. This process was conducted in the local language (Marathi or Hindi, as appropriate) to ensure comprehension. The consent was documented by the research team. Confidentiality of all collected data was strictly maintained, and no personal identifiers were recorded in the analysis or reporting.

## Outcome

This section summarizes the outcomes of the interventions aimed at increasing private TB patients' notifications in Akot Tuberculosis Unit (TU). The findings are presented based on process and outcome measures, analysed during and after the implementation of the PDSA cycles.

### Process measures

The interventions successfully engaged a significant number of private healthcare providers in Akot Tuberculosis Unit (TU).

- **Sensitization Workshops:** Eligible private practitioners, pharmacists, and laboratory technicians were approached through personal visits, follow-up phone calls, and WhatsApp messages explaining the objective of the initiative and inviting them to attend the sessions. Workshops were conducted at the Akot Rural Hospital premises, a centrally located and accessible venue for private providers in the area. A total of three workshops were held over the intervention period (August to October 2023), each lasting approximately two hours. The sessions were facilitated by the DTO and Medical Officers from the district TB cell. The workshop content included an overview of TB notification requirements, a live demonstration of the *Ni-kshay* platform for case reporting, legal implications of non-notification, and hands-on problem-solving of common reporting barriers. Simplified job aids, flowcharts, and contact directories were distributed to all attendees for easy reference. In total, 90 stakeholders including 45 private practitioners, 30 pharmacists, and 15 laboratory technicians attended the sensitization sessions conducted under the intervention. Feedback collected from participants indicated increased awareness, improved understanding of notification procedures, and a greater willingness to participate in TB reporting efforts.

- **Digital Outreach:** WhatsApp messages were sent to private healthcare stakeholders on a weekly basis over a three-month period (August to October 2023), resulting in a total of 50 messages. The messages were designed to be brief, actionable, and motivating, and were delivered through a combination of individual and broadcast communication. The content included simplified steps for using the Ni-kshay portal, reminders about mandatory TB patient notification, motivational quotes, and clarifications on legal obligations and support available from the TB program. The messages were developed in consultation with the District TB Office and tailored to the local context using commonly used terms in Marathi and English. Periodic follow-up messages encouraged two-way communication, allowing stakeholders to ask queries and request additional support when needed.

- **Personalized Follow-Ups:** The follow-up component of the intervention focused on private practitioners who had not previously reported any TB cases on the Ni-kshay platform prior to the intervention. These individuals were identified from the broader pool of private providers mapped within the Akot Tuberculosis Unit and were not limited to stakeholders who attended the sensitization workshops. The Public–Private Mix (PPM) coordinator conducted monthly follow-ups either in person or via telephone during the three-month intervention period. These interactions aimed to resolve practical barriers, including technical difficulties in accessing or navigating the Ni-kshay portal, lack of trained support staff, misconceptions about legal responsibilities, and concerns related to time constraints. By providing personalized assistance and reinforcing key messages, these follow-ups contributed to improved reporting compliance among previously non-notifying practitioners.

### Impact of interventions

The combination of sensitization workshops, digital outreach, and personalized follow-ups the combination of sensitization workshops, digital outreach, and personalized follow-ups contributed to enhanced engagement and compliance among private healthcare providers. Workshop participants reported a clearer understanding of their roles in TB elimination, and

the digital outreach strategy effectively reinforced this knowledge. The regular follow-ups provided a supportive mechanism for overcoming barriers, leading to increased participation in the notification process.

## Results

### Private TB patient notifications

As shown in Table 1, the proportion of TB notifications originating from the private sector increased from 17.8% (16 of 90) during the baseline period (Jan–Jul 2023) to 52.1% (63 of 121) during the post-intervention period (Nov–Dec 2023). A chi-square test demonstrated a statistically significant association between intervention period and private TB notification status ($\chi^2 = 25.91$, $p < 0.001$).

As shown in Table 2, private TB notification performance improved substantially during 2017–2019, reaching 120% of the annual private-sector target in 2019. A marked decline was observed during 2020–2021, coinciding with the COVID-19 pandemic, followed by partial recovery in 2022, which preceded the intervention. Post-intervention years (2023–2024) demonstrated further increases in private TB notifications. These descriptive trends indicate temporal variation influenced by multiple contextual factors rather than a uniform pre-intervention baseline Fig 1. Data for 2024 are presented descriptively to illustrate continuation of post-intervention trends and were not used for causal inference because notification patterns may also reflect concurrent programme changes.

Annual private TB notifications demonstrated a gradual upward trend during the pre-intervention period (2017–2022), although year-to-year variation was observed. Overall, the ITS analysis indicates a statistically significant improvement in post-intervention trends in private TB notifications, without evidence of an immediate level change (Table 3).

**Table 1. Comparison of Private TB Patient Notifications Before and After the PDSA-Based Intervention, Akot TU (2023).**

| Period | Total TB Cases detected | Private TB Notifications (n) | % Private Notifications |
|---|---|---|---|
| Baseline (Jan–Jul 2023) | 90 | 16 | 17.8% |
| Post-intervention (Nov–Dec 2023) | 121 | 63 | 52.1% |

Statistical test: Pearson's chi-square test, $\chi^2 = 25.91$, $p < 0.001$

Source: Ni-kshay portal, Akot TU (2023)

**Table 2. Year-Wise Public and Private TB Patient Notifications Akot Tuberculosis Unit (2017 – 2025*).**

| Year | Public Target | Public Achieved (%) | Private Target | Private Achieved (%) | Total Target | Total Achieved (%) |
|---|---|---|---|---|---|---|
| 2017 | 297 | 180 (61%) | 35 | 8 (23%) | 332 | 188 (57%) |
| 2018 | 347 | 234 (67%) | 35 | 27 (77%) | 382 | 261 (68%) |
| 2019 | 316 | 246 (78%) | 35 | 42 (120%) | 351 | 288 (82%) |
| 2020 | 297 | 136 (46%) | 35 | 31 (89%) | 332 | 167 (50%) |
| 2021 | 290 | 112 (39%) | 35 | 26 (74%) | 325 | 138 (42%) |
| 2022 | 256 | 174 (68%) | 60 | 46 (77%) | 316 | 220 (70%) |
| 2023 | 248 | 132 (53%) | 80 | 79 (99%) | 328 | 211 (64%) |
| 2024 | 240 | 154 (64%) | 80 | 101 (126%) | 320 | 255 (80%) |

*Source: Ni-kshay portal, Akot TU (2023).*

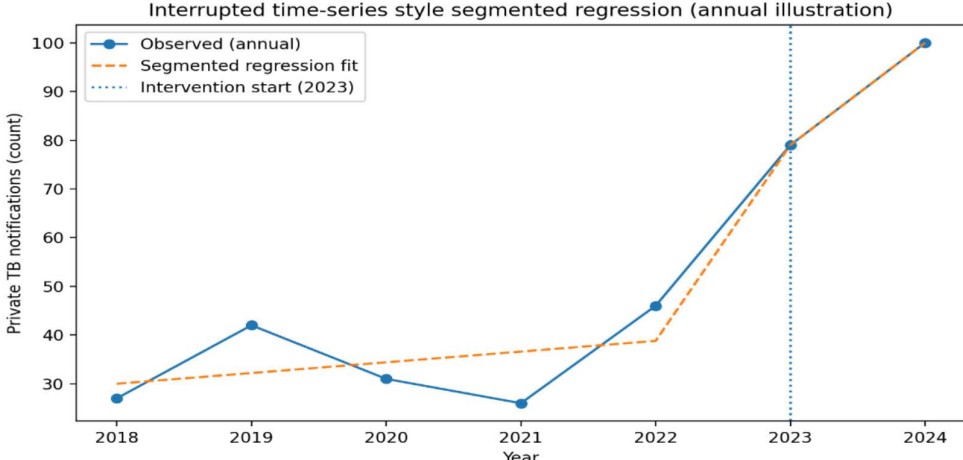

**Fig 1. Observed and *Segmented regression showing annual trend of private TB patient notifications in Akot TU (2018 – 2024).***

**Table 3. Interrupted Time-Series (ITS) Analysis of Private TB Patient Notifications, Akot TU (2018 – 2024).**

| Parameter | Definition | β (Estimate) | 95% CI | p-value | Interpretation |
|---|---|---|---|---|---|
| $\beta_1$ (Time) | Pre-intervention trend (2017–2022) | +5.0 | −2.1 to +12.2 | 0.124 | Gradual but non-significant increase prior to intervention |
| $\beta_2$ (Intervention) | Immediate level change at intervention (2023) | −8.6 | −49.6 to +32.4 | 0.592 | No statistically significant immediate level change |
| $\beta_3$ (Post-trend) | Change in slope after intervention | +57.0 | +13.9 to +100.0 | 0.021 | Significant sustained increase in notification trend post-intervention |

## Qualitative findings: challenges and insights

In addition to quantitative changes in notification patterns, qualitative findings from key informant interviews (KIIs) and field observations provided insights into the contextual, behavioural, and operational factors influencing private-sector TB notification. Thematic analysis identified three interrelated domains: *(1) technical and operational barriers, (2) perceived workload and motivational factors, and (3) the role of sustained support and relationship-building.*

## Technical and operational barriers

Both interview participants and field observations highlighted difficulties in using the Ni-kshay digital platform as a key barrier to notification. Reported challenges included problems with login credentials, uncertainty regarding case submission procedures, incomplete data entry, and limited awareness of platform updates. During field visits, the PPM coordinator observed hesitancy among some providers to independently navigate the platform, particularly in facilities without dedicated data-entry staff. These challenges were addressed through on-site demonstrations, WhatsApp-based video tutorials, and real-time troubleshooting, which were observed to improve provider confidence and consistency in notification practices over successive follow-up visits.

## Perceived workload and motivational factors

KIIs revealed that several practitioners perceived TB notification as an additional administrative burden competing with routine clinical responsibilities, especially in high-volume outpatient settings. Field observations corroborated these

concerns, with providers often prioritizing clinical care during busy hours and deferring notification-related tasks. Through personalized engagement, the PPM coordinator clarified that Ni-kshay case entry required minimal time and facilitated coordination with field staff for partial data completion where needed. Over repeated interactions, observations indicated a gradual reduction in resistance and increased willingness to notify TB cases, particularly when providers perceived tangible programme support.

### Supportive supervision and relationship-building

While digital outreach via WhatsApp was widely appreciated for timely communication, both interviews and observations underscored the importance of periodic in-person follow-ups. Providers frequently expressed greater confidence and motivation when ongoing technical assistance and refresher guidance were available. Field observations documented improved responsiveness to reminders and proactive communication from providers who initially hesitated to engage with notification processes. These findings suggest that sustained; relationship-based supervision played a critical role in reinforcing behavioural change and supporting continued participation in TB notification.

Together, these qualitative findings help contextualize the observed quantitative trends by illustrating how operational barriers, workload perceptions, and supportive engagement mechanisms interacted during implementation. Rather than implying causality, the qualitative data provide explanatory insights into the processes through which the PDSA-based intervention was associated with improved private-sector TB notification.

This flowchart depicts the sequential pathway for TB patient identification and notification through private providers in Akot Tuberculosis Unit (TU), illustrating key operational barriers such as missed referrals, diagnostic delays, and Ni-kshay-related technical gaps that contributed to under-notification prior to the intervention (Fig 2). The visualization highlights critical process points addressed through the PDSA-based quality-improvement cycles to strengthen public–private TB surveillance.

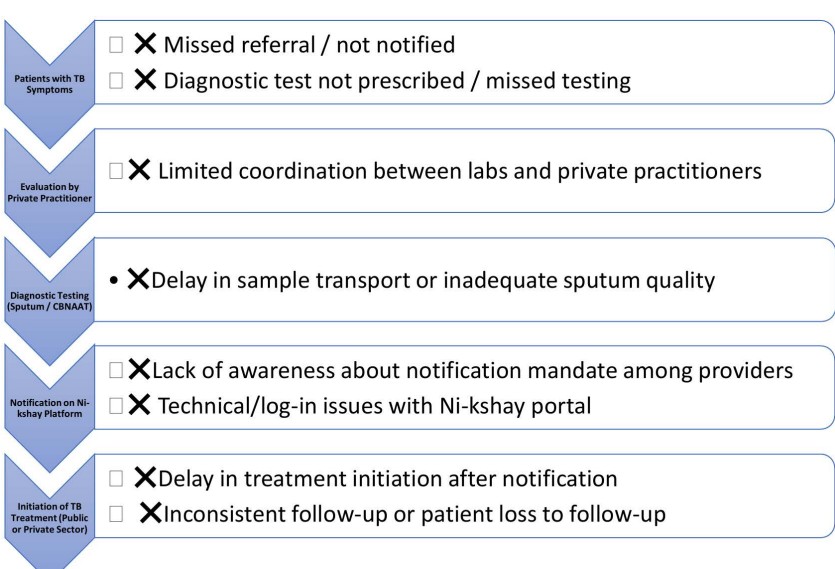

**Fig 2. Patient Flow Pathway and Barriers in TB Notification (Akot TU).**

## Discussion

This study suggests that a systematic and iterative Plan–Do–Study–Act (PDSA) approach was associated with improvements in private-sector engagement in tuberculosis (TB) patient notification systems, addressing a critical operational gap in public health strategies [12]. Through targeted, context-specific interventions, improvements were observed in private TB notifications within the Akot Tuberculosis Unit (TU). These findings contribute insights to the broader discourse on TB elimination efforts and highlight the potential role of structured quality-improvement approaches in strengthening health system processes in resource-limited settings.

The interventions implemented in this study were designed to address locally identified barriers. Sensitization workshops appeared to play an important role in addressing knowledge gaps among private practitioners, pharmacists, and laboratory technicians, particularly regarding the legal and public health importance of TB notification. Digital outreach through WhatsApp achieved high engagement and facilitated timely communication, reinforcement of notification procedures, and clarification of operational issues. These observations are consistent with previous studies highlighting the utility of digital health tools in supporting communication and compliance within TB control programmes in high-burden settings [13].

Personalized follow-ups conducted by the Public–Private Mix (PPM) coordinator were associated with sustained engagement of private providers and helped address practical challenges, including difficulties in using the Ni-kshay platform and misconceptions related to notification responsibilities [14,15]. These findings underscore that technological solutions alone may be insufficient, and that ongoing, relationship-based support mechanisms are important for promoting durable behavioural change among private-sector stakeholders [15,16].

The trends observed in Akot TU should be interpreted alongside broader national and contextual factors. Private-sector contributions to TB notifications in India increased substantially during 2017–2019, followed by a marked decline during 2020–2021, coinciding with the COVID-19 pandemic, and partial recovery in 2022 prior to the intervention. National-level initiatives under the National Tuberculosis Elimination Programme (NTEP), including digital integration through Ni-kshay and incentive-based Public–Private Mix models, have also influenced notification patterns over time [3]. Accordingly, the improvements observed in this study likely reflect the combined influence of local PDSA-based activities, post-pandemic recovery, and ongoing TB programme strengthening efforts.

However, these findings should be interpreted in the context of pre-existing improvements, COVID-19–related disruptions, and concurrent TB programme modifications, rather than being attributed solely to the intervention. The interrupted time-series and trend analyses therefore indicate associations with improved notification patterns rather than definitive causal effects.

Despite these encouraging findings, several limitations should be considered. The study relied on routine programmatic data and self-reported compliance, which may have introduced reporting bias. As the study was conducted in a single tuberculosis unit, the findings may have limited generalizability. In addition, the long-term sustainability of the observed improvements was not assessed. Incomplete month-wise historical notification data limited the ability to conduct detailed time-series analyses, including formal counterfactual projections and assessment of seasonal variation. The observed post-intervention improvements may also reflect the combined influence of the intervention, post-pandemic recovery, and ongoing strengthening of National Tuberculosis Elimination Programme activities rather than the intervention alone. Therefore, the findings should be interpreted as associations rather than definitive causal effects. Further research using longer follow-up periods, complete month-wise data, and multi-site evaluations is needed to better assess the sustainability and impact of PDSA-based interventions on private-sector TB notification.

## Conclusion

This study suggests that context-specific, iterative PDSA-based interventions were associated with improvements in private-sector tuberculosis patient notifications within existing health-system structures. By combining capacity-building,

digital engagement, and supportive supervision, the intervention contributed to addressing technical and behavioural barriers that had previously limited private-provider participation in TB notification. The experience from Akot Tuberculosis Unit suggests that small-scale, data-driven quality-improvement initiatives can complement national strategies, such as Ni-kshay and the National Tuberculosis Elimination Programme, by strengthening implementation at the sub-district level. However, the observed improvements should be interpreted in the context of pre-existing trends, COVID-19–related disruptions, and concurrent TB programme modifications, rather than being attributed solely to the intervention. These findings highlight the potential role of locally adaptable, evidence-based quality-improvement approaches in supporting private-sector engagement in TB surveillance. Further research is needed to assess the sustainability, scalability, and cost-effectiveness of PDSA-based interventions across diverse settings, particularly in high TB-burden countries where private providers remain a critical point of care.

## Supporting information

**S1 File. PDSA cycle documentation and implementation sheets used during the quality-improvement intervention in Akot Tuberculosis Unit, Akola district, Maharashtra.**
(XLSX)

## Acknowledgments

The authors sincerely thank the District Tuberculosis Officer, Akola, and the staff of Akot Tuberculosis Unit for their cooperation and support during the study. We gratefully acknowledge the contributions of private practitioners, pharmacists, and laboratory technicians for their active participation in the PDSA cycles.

## Author contributions

**Conceptualization:** Umesh Kawalkar, Manish Sharma, Abhay Gaidhane, Snehal Chavhan, Mahesh Puri, Priti Kogade.

**Data curation:** Umesh Kawalkar, Manish Sharma, Amar Mankar, Mahesh Puri, Anshu Singh.

**Formal analysis:** Umesh Kawalkar, Manish Sharma, Abhay Gaidhane, Amar Mankar, Snehal Chavhan, Priti Kogade, Anshu Singh.

**Methodology:** Manish Sharma, Amar Mankar, Snehal Chavhan, Mahesh Puri, Anshu Singh.

**Project administration:** Umesh Kawalkar, Manish Sharma, Priti Kogade.

**Resources:** Manish Sharma, Priti Kogade.

**Software:** Snehal Chavhan, Mahesh Puri, Anshu Singh.

**Supervision:** Umesh Kawalkar, Manish Sharma, Abhay Gaidhane.

**Validation:** Umesh Kawalkar, Abhay Gaidhane, Zahiruddin Quazi Syed.

**Writing – original draft:** Umesh Kawalkar, Amar Mankar.

**Writing – review & editing:** Manish Sharma, Abhay Gaidhane, Snehal Chavhan, Mahesh Puri, Priti Kogade, Anshu Singh, Zahiruddin Quazi Syed.

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
