## [Decision Letter · Decision Letter 0]

4 Sep 2025

PGPH-D-25-01093

Improving Private TB Notifications in Akot Tuberculosis Unit of Vidarbha region of India: A Plan-Do-Study-Act (PDSA) Based Approach for Enhanced Public Health Reporting

Dear Dr. Mankar,

Thank you for submitting your manuscript to PLOS Global Public Health. After careful consideration, we feel that it has merit but does not fully meet PLOS Global Public Health’s publication criteria as it currently stands. Therefore, we invite you to submit a revised version of the manuscript that addresses the points raised during the review process.

The manuscript has been evaluated by three reviewers, and their comments are available below. Could you please carefully revise the manuscript to address all comments raised?

The reviewers raised significant concerns with the analysis and reporting. Please ensure you address Reviewer 3's comments relating to the statistical methods used and whether they are appropriate for the type of data in question. Please also update the results so that they are reported with an appropriate level of detail, as per the comments of Reviewers 1 and 2.

Thank you.

We look forward to receiving your revised manuscript.

Kind regards,

Alejandro Torrado Pacheco, PhD

Staff Editor

Journal Requirements:

1. In the online submission form, you indicated that “The data supporting the findings of this study are available from the corresponding author upon reasonable request. All data were collected in accordance with ethical standards, and participant confidentiality has been maintained.”.

a. In a public repository,

b. Within the manuscript itself, or

c. Uploaded as supplementary information.

2. We have amended your Competing Interest statement to comply with journal style. We kindly ask that you double check the statement and let us know if anything is incorrect.

3. Please provide a/amend your detailed Financial Disclosure statement. This is published with the article. It must therefore be completed in full sentences and contain the exact wording you wish to be published.

**Please only choose the relevant sentences from below**

a. Please clarify all sources of funding (financial or material support) for your study. List the grants (with grant number) or organizations (with url) that supported your study, including funding received from your institution.

b. State the initials, alongside each funding source, of each author to receive each grant.

c. State what role the funders took in the study. If the funders had no role in your study, please state: “The funders had no role in study design, data collection and analysis, decision to publish, or preparation of the manuscript.”

d. If any authors received a salary from any of your funders, please state which authors and which funders.

Additional Editor Comments (if provided):

Reviewers' comments:

Reviewer's Responses to Questions

**Comments to the Author**

1. Does this manuscript meet PLOS Global Public Health’s publication criteria? Is the manuscript technically sound, and do the data support the conclusions? The manuscript must describe methodologically and ethically rigorous research with conclusions that are appropriately drawn based on the data presented.

Reviewer #1: Partly

Reviewer #2: Yes

Reviewer #3: No

2. Has the statistical analysis been performed appropriately and rigorously?

Reviewer #1: Yes

Reviewer #2: Yes

Reviewer #3: No

3. Have the authors made all data underlying the findings in their manuscript fully available (please refer to the Data Availability Statement at the start of the manuscript PDF file)?

Reviewer #1: Yes

Reviewer #2: Yes

Reviewer #3: No

4. Is the manuscript presented in an intelligible fashion and written in standard English?

Reviewer #1: Yes

Reviewer #2: Yes

Reviewer #3: No

5. Review Comments to the Author

Reviewer #1: Thank you for focusing on this important topic. The findings have potential to improve private provider notifications in other settings inside and outside of India. Here are some suggestions to strengthen the impact of your message:

- please consider reporting the annual TB case notification for Akot TU. This will help readers get a sense of the TB incidence of this area, as well as the proportion of cases that are currently reported by private providers. (e.g. 16 of 32 total TB cases vs. 16 of 500...)

-Reporting both the number and proportion of TB case notifications that are from private providers will allow standardization (in years where there are a lot of TB notifications, the increase of private provider notification may be just a proportional increase, vs. a true increase in pp notification resulting from the intervention.

Please consider reporting the change in TB notifications as changes in proportion of notifications from pp as a fraction of all TB notifications.

line 46: please provide an estimate of the size of private provider network in the area so we have a sense of the denominator for the folks that received the intervention

line 46: missing a word to go with 16 (persons with TB? notifications? providers?)

line 141: need to capitalize the first word

graph 2: per what is reported, there were 16+63, or 79 TB notifications from private providers. The graph states that there were 43 for that year. You'll need to figure out which is correct.

If there were 79 notifications from 2023, the segmented regression needs to be recalculated. There might be a statistical difference now.

Figure 4: I would suggest separating this out into 2 figures: patient flow diagram and text box with identified gaps.

Patient flow diagram: You can merge the 4 balloons in the upper left into one box of possible ways patients come to pp. Each balloon can become one bullet.

The barriers can be signified by something that blocks the flow of the arrows. There are other published articles you may want to look at. Currently, these boxes use thicker arrows, which look like inputs, not barriers.

You may consider having a separate text box that bullets out the major gaps that were identified in your PDSA cycle. These are really important for readers from other settings that might have similar issues.

The wording for the flow diagram and list of gaps can be made more concise

in your flow diagram you have TB treatment by pp as one of the end outcomes. Were you able to assess whether they were on the correct treatment or not? If you were, this would be something interesting to report out.

Reviewer #2: Authors have summarized the intervention in the manuscript along with appropriate analysis. The results appear encouraging enough to be reported scientifically. However, there are some internal consistency issues as noticed by the reviewer as mentioned below:

Authors mention in line #40-41 that - private notifications accounted for only 32% of TB cases in 2017 but increased to 90% by 2023. However, it is unclear for the readers as to what exactly is the percentage. Is the private notification out of total? Or is it out of the target set? Or is it out of any other available denominator? Also, the percentage appears to be increasing to 93% in 2023 itself when the annual increase was non-significant @4% annually mentioned later in the manuscript. Significant increase in private sector notification actually happened in 2024. Then did the percentage go above 100% in 2024? These questions are likely for the reader. Setting the right context is important, so that the results of the intervention can be interpreted rightfully. Authors are encouraged and advised to provide details of the same to demystify the percentage of private sector notification.

Authors, upfront, in the abstract mentions that – ‘quality improvement study was conducted’, however as per objective ‘To implement and evaluate a Plan-Do-Study-Act (PDSA)-based intervention to enhance private sector TB notification rates’, whole analysis is based on the quantitative difference measurement before and after the intervention in the private sector notification. Though it is mentioned that the qualitative assessment was done using KIIs and FGDs, it is also mentioned that the findings were helpful in triangulation. However, there is no mention of such important findings and the analysis or testing of the hypothesis that the quantitative improvement in notification happened due to the qualitative methods in interventions as a result of KIIs. Authors are encouraged to provide such details of qualitative findings and its correlation with the study findings.

Authors have submitted the manuscript to the global public health journal which has global audience. Considering this, authors are advised to explain the terminologies like Nikshay. This will help in clarity for the readers and in setting the context better. Also, it is understood by reviewers that the actual name of the digital platform is ‘Ni-kshay’ and not ‘Nikshay’. Authors are suggested to make the corrections accordingly in the whole manuscript and also explain the meaning of the word ‘Ni-kshay’

Authors repeatedly mention that only 16 cases were notified from private healthcare providers during the baseline period. However, there is no clarity on what was the universe? How many health facilities actually existed in the study area? What are different sources of such information? Why was only Ni-kshay used as a source of information for the list of health facilities? What was the extent of coverage or completeness of inclusion of health facilities in Ni-kshay? How many facilities were outside Ni-ksahy? How were these ascertained (source of alternative information e.g. line list available with District Health Officer office, clinical establishment registry, Medical Associations, private directories like yellow pages, practo etc) and basis of not including them in the study. These issues seriously challenge the validity of the study due to potential selection bias. Authors are encouraged to explain such details to ensure scientific credibility of the article.

It is not understood as to why, the stakeholders who had already been regularly notifying TB cases, were excluded from the study? These stakeholders could have high chances of large cases being diagnosed or managed by them and reporting only few with high under-reporting. Such observations have been reported by several studies. In short, what was the basis for differentiating those under-reporting with those who are complete reporting? If it is only the comparison with previous notification by them, it may mislead. Authors may like to explain more in details about their rationale behind the decision of such exclusion.

Authors are encouraged to add a flow diagram of the process with details of types of stakeholders and their total numbers in the study area, what numbers / proportion were available in Ni-kshay, numbers excluded with reasons, number included with types, numbers interviewed & numbers who could not be interviewed, numbers who were followed up for each month etc.

It appears that authors have not followed the basic norm for providing the captions for graphs for example graph 1 and graph 2 do not have title. Authors are encouraged to read the journal’s authors guideline for preparation and submission of manuscript and follow the same.

Right from title and though out the manuscript the authors have mentioned TB notification. While it is a convenient interpretation that TB notification means notification of TB patient; it is not the only case. Any death due to tuberculosis or otherwise is also notifiable. Similarly, the TB Preventive Treatment and eligibility of all contacts of TB patients are also required to be notified as a public health action by the health facilities. Considering this diversity in the meaning of the word notification, authors are advised to make change as TB patient notification throughout the manuscript to avoid any erroneous interpretation.

Reviewer #3: The authors present an intervention to improve private sector notification in an Indian district. While I appreciate the intervention and can see the effort that must have went into this intervention, significant analytical flaws prevent acceptance in the current form.

Major issues with Interrupted Time Series (ITS) analysis:

1. Insufficient post-intervention periods: The authors use only one post-intervention time point, which is inadequate for ITS regression analysis. Multiple post-intervention periods are essential for proper trend assessment. A difference-in-difference approach with explicit time factors may be more appropriate for this data structure.

2. Inappropriate distribution: The authors appear to use Gaussian regression for count data. Poisson or negative binomial regression with population offset would be more suitable for this data type.

3. Model specification unclear: The regression parameters lack clarity. Including the actual model equation rather than describing it as "segmented regression" would improve transparency and reproducibility.

4. Temporal reporting inconsistencies: The description contains multiple conflicting time periods that create confusion about the study timeline.

These methodological concerns significantly impact the validity of the conclusions drawn.

6. PLOS authors have the option to publish the peer review history of their article (what does this mean?). If published, this will include your full peer review and any attached files.

**Do you want your identity to be public for this peer review?** For information about this choice, including consent withdrawal, please see our Privacy Policy.

Reviewer #1: No

Reviewer #2: **Yes:** Kirankumar Rade

Reviewer #3: No

---

## [Decision Letter · Decision Letter 1]

23 Nov 2025

PGPH-D-25-01093R1

Improving Private Tuberculosis Patient Notifications in Akot Tuberculosis Unit of Vidarbha region of India: A Plan-Do-Study-Act (PDSA) Based Approach for Enhanced Public Health Reporting

Dear Amar Mankar,

Thank you for submitting your manuscript to PLOS Global Public Health. After careful consideration, we feel that it has merit but does not fully meet PLOS Global Public Health’s publication criteria as it currently stands. Therefore, we invite you to submit a revised version of the manuscript that addresses the points raised during the review process.

While all reviewers acknowledge the major work done to revise the manuscipt following the first round of comments, there are still outstanding issues. A major concern remains the statistical methods and results presented. Kindly address the comments from reviewer 1 and 3 in your next submission.

We look forward to receiving your revised manuscript.

Kind regards,

Indira Govender

Academic Editor

Journal Requirements:

1. Please provide separate figure files in .tif or .eps format only and remove any figures embedded in your manuscript file. Please also ensure that all files are under our size limit of 10MB. Please leave the figure captions in the manuscript.

Additional Editor Comments (if provided):

Reviewers' comments:

Reviewer's Responses to Questions

**Comments to the Author**

1. If the authors have adequately addressed your comments raised in a previous round of review and you feel that this manuscript is now acceptable for publication, you may indicate that here to bypass the “Comments to the Author” section, enter your conflict of interest statement in the “Confidential to Editor” section, and submit your "Accept" recommendation.

Reviewer #1: (No Response)

Reviewer #2: All comments have been addressed

Reviewer #3: (No Response)

2. Does this manuscript meet PLOS Global Public Health’s publication criteria? Is the manuscript technically sound, and do the data support the conclusions? The manuscript must describe methodologically and ethically rigorous research with conclusions that are appropriately drawn based on the data presented.

Reviewer #1: Partly

Reviewer #2: Yes

Reviewer #3: No

3. Has the statistical analysis been performed appropriately and rigorously?

Reviewer #1: No

Reviewer #2: Yes

Reviewer #3: No

4. Have the authors made all data underlying the findings in their manuscript fully available (please refer to the Data Availability Statement at the start of the manuscript PDF file)?

Reviewer #1: Yes

Reviewer #2: Yes

Reviewer #3: (No Response)

5. Is the manuscript presented in an intelligible fashion and written in standard English?

Reviewer #1: Yes

Reviewer #2: Yes

Reviewer #3: Yes

6. Review Comments to the Author

Reviewer #1: The revised manuscript is much improved, but there are a few final issues that need to be corrected.

line 47 incorrectly states that 90% of the reported TB notifications were from the private sector. The 2024 India report shows 0.84 lakh from the private sector of 2.55 lakh total. That is ~33%. The 2023 report shows 7.4 lakh (30%) of 24.2 lakh total TB notifications. Please correct this number as it is used throughout the paper. TB notifications from the private sector have been about 1/3 of all notifications. Your intervention is still noteworthy and successful.

line 237: What was the total pool of low-reporting private providers? This would provide the denominator for the 90 that attended the sensitization. I hope I am interpreting the activity correctly.

line 259: there is a slight inaccuracy in using the 211 total number of cases as the denominator for both the pre-intervention (Jan-July) and intervention (Aug-Dec) periods. The 16 cases should be divided by those cases diagnosed from Jan-July and the 63 cases by the number cases diagnosed between Aug - Dec. As TB diagnoses are not evenly distributed throughout the year, this would provide a more accurate proportion of private TB notifications during these two different periods. It may actually increase the % during the intervention period.

Thank you for your hard work on this!

Reviewer #2: Comments have been addressed by authors in the revision.

Reviewer #3: I appreciate the authors' efforts to address the previous concerns. However, significant methodological and analytical issues remain that preclude acceptance.

1. Interrupted Time Series (ITS) Analysis

The ITS implementation continues to have fundamental problems:

a. The authors state they used monthly notification data but present results annually. If the intervention start point falls mid-year and monthly data are available, monthly results should be presented to properly implement ITS.

b. Population offset concerns:

• It is not clear if the authors used the same population for the entire time period or were annual population projections used.

• Poisson distribution with population offset results reflect the rate change per unit population. I find it hard to believe that there was an increase of 50.5 cases per unit population post-intervention. The authors present no units for coefficients and do not clarify whether these are crude log coefficients or exponentiated values. Please present results with appropriate units and interpretation.

• The model specification is still not clear. How did you code the post-trend? The values seem very high for a slope change. Please check https://doi.org/10.1093/ije/dyaa148 for how to parametrize and interpret ITS models.

c. The COVID-19 pandemic effect has not been appropriately incorporated.

d. With monthly data, seasonal patterns should be examined.

2. Chi-Squared Analysis (Table 1)

a. Interpretation error (lines 261-263): The interpretation of chi-squared is incorrect. The authors are conflating between association and causation. Statistical significance does not mean there is causation.

b. Computational errors: I am unable to reproduce your chi-squared results. Please see the R code and results below:

> # Scenario 1

> tab <- matrix(c(16, 195,

+ 63, 148),

+ nrow = 2, byrow = TRUE)

> chisq.test(tab)

Pearson's Chi-squared test with Yates' continuity correction

data: tab

X-squared = 32.954, df = 1, p-value = 0.000000009437

> chisq.test(tab, correct = FALSE)

Pearson's Chi-squared test

data: tab

X-squared = 34.402, df = 1, p-value = 0.000000004482 >

> # Scenario 2

> tab2 <- matrix(c(16, 211,

+ 63, 211),

+ nrow = 2, byrow = TRUE)

> chisq.test(tab2)

Pearson's Chi-squared test with Yates' continuity correction

data: tab2

X-squared = 22.577, df = 1, p-value = 0.000002019

> chisq.test(tab2, correct = FALSE)

Pearson's Chi-squared test

data: tab2

X-squared = 23.762, df = 1, p-value = 0.00000109

It seems that the authors have not used the correct denominator. 211 is the annual notification. Please recheck the table and presentation of the results.

3. Trend Interpretation (Table 2)

The trends from Table 2 need to be interpreted in a better manner. Looking at the numbers, I see that there was an increase in the target achievement from 2017-2019. Private sector notification was 120% of the target in 2019. The drop in 2020 and 2021 might be attributed to the COVID-19 pandemic (which has not been included in the ITS model. If the authors have monthly data I suggest to do at least sensitivity analysis for this). In 2022, which is pre-intervention period, we see an uptick in the notification.

Attributing notification changes solely to this intervention ignores substantial contextual factors, including the pandemic and other TB program modifications over time. The crude trend analysis cannot support causal claims. Please substantially moderate your language.

4. Qualitative Methods

Critical methodological details are missing:

• Who collected the data and when?

• How were interview participants selected?

• Who were the key informant interviewees?

• What observation data were collected?

Qualitative results only mention challenges and mitigation, but report nothing from observations.

Additional details are necessary in the methods and results sections for the qualitative part.

5. Manuscript Organization

Overall, the organization of the manuscript also leaves a lot to be desired. There are multiple places in the manuscript where methods, results and discussions are mixed in the inappropriate sections. For example, lines 246-252 seems like something for the results section. Lines 271-273 are discussing the results.

While I acknowledge the authors' work on this intervention, the analytical quality and manuscript presentation do not meet publication standards and major revision is required.

7. PLOS authors have the option to publish the peer review history of their article (what does this mean?). If published, this will include your full peer review and any attached files.

**Do you want your identity to be public for this peer review?** For information about this choice, including consent withdrawal, please see our Privacy Policy.

Reviewer #1: No

Reviewer #2: **Yes:** Kiran Rade

Reviewer #3: No

Figure Resubmissions:

---

## [Decision Letter · Decision Letter 2]

9 Feb 2026

PGPH-D-25-01093R2

Improving Private Tuberculosis Patient Notifications in Akot Tuberculosis Unit of Vidarbha region of India: A Plan-Do-Study-Act (PDSA) Based Approach for Enhanced Public Health Reporting

Dear Dr. Mankar,

Thank you for submitting your manuscript to PLOS Global Public Health. After careful consideration, we feel that it has merit but does not fully meet PLOS Global Public Health’s publication criteria as it currently stands. Therefore, we invite you to submit a revised version of the manuscript that addresses the points raised during the review process.

EDITOR:

The reviewers have still raised concerns about your analytic method. Please address these. If necessary, but consulting an expert. If (after that consultation) you cannot address the challenges given the available data, then this needs to be described in detail as a limitation to extrapolating from your findings.

We look forward to receiving your revised manuscript.

Kind regards,

Graeme Hoddinott, Ph.D

Academic Editor

Journal Requirements:

Additional Editor Comments (if provided):

Reviewers' comments:

Reviewer's Responses to Questions

**Comments to the Author**

1. If the authors have adequately addressed your comments raised in a previous round of review and you feel that this manuscript is now acceptable for publication, you may indicate that here to bypass the “Comments to the Author” section, enter your conflict of interest statement in the “Confidential to Editor” section, and submit your "Accept" recommendation.

Reviewer #1: (No Response)

Reviewer #2: All comments have been addressed

2. Does this manuscript meet PLOS Global Public Health’s publication criteria? Is the manuscript technically sound, and do the data support the conclusions? The manuscript must describe methodologically and ethically rigorous research with conclusions that are appropriately drawn based on the data presented.

Reviewer #1: Partly

Reviewer #2: Yes

3. Has the statistical analysis been performed appropriately and rigorously?

Reviewer #1: I don't know

Reviewer #2: Yes

4. Have the authors made all data underlying the findings in their manuscript fully available (please refer to the Data Availability Statement at the start of the manuscript PDF file)?

Reviewer #1: Yes

Reviewer #2: Yes

5. Is the manuscript presented in an intelligible fashion and written in standard English?

Reviewer #1: Yes

Reviewer #2: Yes

6. Review Comments to the Author

Reviewer #1: Congratulations on your improved paper; the hard work you have put in shows. The paper reads much clearer and the impact of your intervention is also more persuasively laid out.

I still wonder about why you chose the first 7 months of 2023 as baseline? Without the intervention would the 2023 counts have been projected to be 60 or would it have been higher? Why not include 2022 to stabilize your baseline? I am not an expert in this methodology, but I still have some questions on this, given that only annual numbers are available. Also, any possibility of including private notifications from 2024?

Other than that, I think the paper provides TB programs at local and district levels a feasible way to engage private providers and improve notification.

Reviewer #2: All comments are responded and amendments are made in manuscript by authors

7. PLOS authors have the option to publish the peer review history of their article (what does this mean?). If published, this will include your full peer review and any attached files.

**Do you want your identity to be public for this peer review?** For information about this choice, including consent withdrawal, please see our Privacy Policy.

Reviewer #1: No

Reviewer #2: No

 Figure Resubmissions:

---

## [Decision Letter · Decision Letter 3]

23 Feb 2026

PGPH-D-25-01093R3

Improving Private Tuberculosis Patient Notifications in Akot Tuberculosis Unit of Vidarbha region of India: A Plan-Do-Study-Act (PDSA) Based Approach for Enhanced Public Health Reporting

Dear Dr. Kawalkar,

Thank you for submitting your manuscript to PLOS Global Public Health. After careful consideration, we feel that it has merit but does not fully meet PLOS Global Public Health’s publication criteria as it currently stands. Therefore, we invite you to submit a revised version of the manuscript that addresses the points raised during the review process.

EDITOR:

We have reached somewhat of an impasse with this submission. Two previous reviewers (before I was editor) raised significant concerns about the methods / ITS-approach. They are no longer available to review. Two other reviewers (not method experts) were generally happy with the content / topic, but unsure about the methods. A further reviewer specifically invited as a methods expert (and reading only the most recent version) has also raised significant concerns. My view is that these are substantive and requires you to re-think and re-run the analysis before revision. This may not be possible. If so, then unfortunately we will not be able to accept this submission. Please consider the most recent reviewer comments very closely.

We look forward to receiving your revised manuscript.

Kind regards,

Graeme Hoddinott, Ph.D

Academic Editor

Journal Requirements:

Additional Editor Comments (if provided):

Reviewers' comments:

Reviewer's Responses to Questions

**Comments to the Author**

1. If the authors have adequately addressed your comments raised in a previous round of review and you feel that this manuscript is now acceptable for publication, you may indicate that here to bypass the “Comments to the Author” section, enter your conflict of interest statement in the “Confidential to Editor” section, and submit your "Accept" recommendation.

Reviewer #4: (No Response)

2. Does this manuscript meet PLOS Global Public Health’s publication criteria? Is the manuscript technically sound, and do the data support the conclusions? The manuscript must describe methodologically and ethically rigorous research with conclusions that are appropriately drawn based on the data presented.

Reviewer #4: No

3. Has the statistical analysis been performed appropriately and rigorously?

Reviewer #4: No

4. Have the authors made all data underlying the findings in their manuscript fully available (please refer to the Data Availability Statement at the start of the manuscript PDF file)?

Reviewer #4: Yes

5. Is the manuscript presented in an intelligible fashion and written in standard English?

Reviewer #4: Yes

6. Review Comments to the Author

Reviewer #4: My review is focused on the Methods as requested by the Editor. There are a number of issues that I feel need to be addressed that preclude publication of the manuscript in its current format.

1. The authors note that “Stakeholders (n: 106) who had already been regularly notifying TB cases were excluded to focus efforts on those with the 107 highest potential for change.” Are the TB notification data reported then limited to the stakeholders included, or do they reflect data from all private practitioners? If the latter as I presume, it should be clarified that the intervention was targeted to certain stakeholders, but data from all stakeholders were included in the analysis.

2. The intervention included linking private providers to Nikshay who were not previously linked. The pre- and post-intervention periods therefore include different sets of providers who could report TB notifications. And the post-intervention period includes a much higher number of linked providers. This on its own makes it likely that a higher number of notifications would be seen in the post-intervention period.

3. It seems that 90 TB notifications were reported in the 7-month (Jan – July 2023) pre-intervention period, but 121 cases were notified during the 2-month post-intervention period (Nov-Dec 2023). This suggests there are temporal trends in case notifications, and that the pre- and post-intervention periods should cover the same calendar periods. Could the authors report case notifications for Nov 2022 – July 2023 vs. Nov 2023 – July 2024 instead of Jan-July 2023 vs. Nov-Dec 2023?

4. The ITS analysis time periods do not make sense. The count of private TB notifications is very low from 2018 to 2022, jumps markedly between 2022 and 2023 and continues to increase from 2023 to 2024 (but with minimal change in upward trend from 2022-2024). It therefore does not make sense to include the 2018-2022 data. I would focus on monthly (or bi-monthly) time periods from 2022 to 2024 and then assess trends. Ideally, the ITS should cover the same time period as suggested in comment 3 above: pre-intervention period Nov 2022 – July 2023 vs. post-intervention period Nov 2023 – July 2024). This facilitates comparison of the most proximate pre- and post-intervention periods, and allows an assessment of whether findings from the simple pre-post comparison referenced in comment 3 are different from when using a more sophisticated analytic technique such as ITS.

7. PLOS authors have the option to publish the peer review history of their article (what does this mean?). If published, this will include your full peer review and any attached files.

**Do you want your identity to be public for this peer review?** For information about this choice, including consent withdrawal, please see our Privacy Policy.

Reviewer #4: No

Figure Resubmissions:

---

## [Editor Report · Decision Letter 4]

30 Apr 2026

Improving Private Tuberculosis Patient Notifications in Akot Tuberculosis Unit of Vidarbha region of India: A Plan-Do-Study-Act (PDSA) Based Approach for Enhanced Public Health Reporting

PGPH-D-25-01093R4

Dear Dr Kawalkar,

We are pleased to inform you that your manuscript 'Improving Private Tuberculosis Patient Notifications in Akot Tuberculosis Unit of Vidarbha region of India: A Plan-Do-Study-Act (PDSA) Based Approach for Enhanced Public Health Reporting' has been provisionally accepted for publication in PLOS Global Public Health.

Best regards,

Graeme Hoddinott, Ph.D

Academic Editor

Reviewer Comments (if any, and for reference):

Please consider further strengthening the paragraph on 'limitations' to more explicitly address the challenges of an ITS analysis when the time intervals are so broad and what problems of mis- or over-interpretation this may lead to.